physical chemistry

amino acid, colloids, catalysis, gemini surfactant, kinetics, interfacial properties

**Author for correspondence:**
Dileep Kumar
e-mail: dileepkumar@tdtu.edu.vn

This article has been edited by the Royal Society of Chemistry, including the commissioning, peer review process and editorial aspects up to the point of acceptance.

# Synthesis and characterization of geminis and implications of their micellar solution on ninhydrin and metal amino acid complex

Dileep Kumar[1,2], Malik Abdul Rub[3,4]
and Abdullah M. Asiri[3,4]

[1]Division of Computational Physics, Institute for Computational Science, Ton Duc Thang University, Ho Chi Minh City, Vietnam
[2]Faculty of Applied Sciences, Ton Duc Thang University, Ho Chi Minh City, Vietnam
[3]Chemistry Department, Faculty of Science, King Abdulaziz University, Jeddah 21589, Saudi Arabia
[4]Center of Excellence for Advanced Materials Research, King Abdulaziz University, Jeddah 21589, Saudi Arabia

DK, 0000-0003-2913-5032

In our study, three gemini dicationic surfactants with different methylene group spacer (16-6-16, 16-5-16 and 16-4-16) have been synthesized and characterized in solution by $^1H$ NMR spectroscopic technique. The implications of gemini micellar solution on ninhydrin and metal amino acid complex ($[Cu(II)\text{-}Trp]^+$) were performed by the means of single-beam UV–visible spectroscopy. The absorbance was noted at regular time intervals and values of rate constant ($k_\psi$) were determined by using a computer-based program. Synthesized surfactants proved as an efficient catalyst on the interaction of ninhydrin with metal amino acid complex as compared with conventional surfactant and aqueous systems. The required description regarding the implications of gemini dicationic surfactants are provided in the text in detail. The conductivity technique was applied in order to get critical micelle concentration (cmc) of geminis in the presence and absence of reactants. Catalytic results developed in gemini dicationic surfactant system were explained effectively by pseudo-phase model. Various thermodynamic quantities, *viz.*, activation energy, $E_a$, activation enthalpy, $\Delta H^\#$, and activation entropy, $\Delta S^\#$, were obtained on interaction of ninhydrin with $[Cu(II)\text{-}Trp]^+$ in gemini systems by applying Eyring equation. A detailed explanation about these evaluated parameters was also made.

# 1. Introduction

Surfactant, referred to as a surface-active material, is capable of reducing interfacial tension. They have polar and non-polar moieties called head group and hydrophobic tail, respectively. Their strength of interaction depends significantly on the nature of hydrophilic heads and hydrophobic tail. So, surfactants have been used in various industrial applications, such as catalyst, cosmetics, oil exploration, daily chemical and pharmaceutical needs, textile industries, dying and painting [1–8]. Commonly, the efficacies of surfactants in uses depend upon critical concentration known as concentration (cmc). It is defined as a minimum range of concentration at which surfactant monomer initiates to self-associate that can be obtained as an inflection point by plotting any of the physico-chemical properties against surfactant concentration [9–14]. Surfactants self-aggregate and turn into several morphological aggregates above cmc, e.g. bilayers, vesicles, micelles and nanostructures. Therefore, surfactants play several important roles in diverse physico-chemical properties [15,16].

Gemini quaternary ammonium surfactants, attached by two hydrophilic head groups and two hydrophobic chains at or close to heads through a spacer, have received the great consideration by several investigators for their outstanding features [17]. Both the hydrophilic head and hydrophobic chain have special chemical structures and are responsible for their excellent uses in numerous purposes (industrial and commercial applications) [18]. Gemini surfactants are a unique class of surfactants and there are great significances for their excellent interfacial properties [19]. In contrast to conventional monoalkyl cationic surfactants, they have a wide range of chemical and structural morphologies and exhibit properties such as good viscoelasticity, better solubilizing capacity, low cmc value, excellent wettability and so on [20–25].

Being dependent on alkyl head group, hydrophobic chain and length of spacer as well as structures of consisting species, studies of gemini surfactants and their aggregates provide several valuable applications in the field of surface and interfacial sciences [26–30]. Most of the current reports available are also focused on the micellar and surface-active properties of gemini surfactants [31–34]. Authors have investigated and reported that gemini surfactants were found to be superior as drug delivery agents in medical and pharmaceutical sciences [35–38]. Even though a large number of scientific reports are existing on surface behaviour of gemini and their aggregates that they form, the reports on the studies of their influences on rates have not received the considerable attraction. However, the complexity in the synthesizing and purifying of gemini surfactants hinders the usage and applications in most domestic and industrial areas.

Therefore, in order to fulfil the growing requirements of several industries and commercial utilization, three dicationic gemini surfactants having various methylene spacer chain length (e.g. 16-6-16, 16-5-16 and 16-4-16) have been synthesized and characterized by using $^1$H NMR spectroscopy. Influences of these synthesized gemini materials on the rates of interaction of ninhydrin with metal amino acid complex have been studied in sufficient manner. We believe that the outcome of the present study will increase the awareness in regard of the use of gemini surfactants and will expand their scope of application to large scale. The findings of study in gemini are also compared with that obtained in aqueous system.

Ninhydrin, an effective colour-generating chemical compound, is used largely to classify the amine functional group in the several domains, e.g. biochemical studies, chemical works and forensics [39,40]. Studies of interaction relating ninhydrin with amine functional group offer a number of biological significances related to living organism (such as, transpeptidation and deamination) [41,42]. Reaction of ninhydrin and amino group yields the diketohydrindylidene-diketohydrindamine (DYDA) commonly called to Ruhemann's purple. As the DYDA destabilizes at room temperature, many developments (e.g. effect of traditional monoalkyl surfactants, role of various salts, impact of different solvent media, and so on) were made to stabilize the DYDA [43–47]. Whereas, effects of gemini surfactants on amino group and ninhydrin are scanty and have not obtained essential attention. Investigators/scientists working in similar or allied arenas are still awaiting the better outcomes and significances.

# 2. Experimental section

## 2.1. Materials and methods

All the materials applied in the present work is listed in table 1.

**Table 1.** Source and purity of materials applied in present work.

| name of the materials | source | purity in mass fraction | CAS/batch/ lot number | purification methods | analysis method |
|---|---|---|---|---|---|
| CH₃COOH | Merck (India) | 0.99 | HH3H530442 | none | none |
| CH₃COONa | Merck (India) | 0.99 | ML0M603893 | none | none |
| copper sulfate | Merck (India) | 0.98 | 7758-98-7 | vacuum drying | none |
| 1,6-dibromohexane | Fluka (Germany) | 0.97 | 629-03-8 | vacuum drying | none |
| 1,5-dibromopentane | Fluka (Germany) | 0.98 | 111-24-0 | vacuum drying | none |
| 1,4-dibromobutane | Fluka (Germany) | 0.98 | 110-52-1 | vacuum drying | none |
| *N,N*-dimethylcetylamine | Fluka (Germany) | 0.95 | 112-69-6 | vacuum drying | none |
| DL-tryptophan | SRL (India) | 0.99 | 71-00-1 | vacuum drying | none |
| ninhydrin | Merck (India) | 0.99 | DC2DR52232 | none | none |
| ethyl acetate | Merck (India) | 0.99 | IK0IF60606 | none | none |
| ethanol absolute | Merck (Germany) | 0.998 | K40488983944 | none | none |

Gemini dicationic surfactants (16-6-16, 16-5-16 and 16-4-16) were synthesized in the laboratory and the detailed methods were mentioned in the published articles [48,49]. Synthesized surfactants were characterized by $^1$H NMR technique and were matched in close agreement to results reported formerly [48,49]. Water used throughout the experiment was demineralized double-distilled from alkaline $KMnO_4$. The specific electrical conductivity of water employed was 1–2 µs cm$^{-1}$ at 298 K. Standard solutions of complex, ninhydrin and surfactants were made by accurate weighing of required quantity using an acetate buffers. All the solutions were stirred well to be homogenized and kept for a day to attain equilibrium at room temperature. To measure the solution of pH, a digital Elico pH meter (Hyderabad, India) was used.

## 2.2. Electrical conductivity measurements

Electrical conductivities were measured on conductivity meter (Systronics model 306, Ahmedabad, India) in order to get cmc at required experimental temperatures (i.e. 303 K and 343 K). Solutions of gemini and the mixed additives were left at room temperature to ensure stabilization. For cmc evaluation, [ninhydrin] and [complex] were fixed at 6 and 0.2 mmol kg$^{-1}$, respectively. Each run was repeated at least in triplicate to get reproducible results. Before starting the study, apparatus was calibrated with a solution of potassium chloride at different concentrations. For determining cmc, specific conductivities were plotted against different concentrations of gemini surfactants and the inflection point in the plot corresponds to the cmc value [50–55]. An effective enhancement in conductivity was noted in premicellar region owing to free cations and anions but not in post region due to formation of micelle. In our study, cmc of pure gemini obtained is consistent at 303 K with outcomes published formerly [56]. The cmc values at various reaction situations (i.e. water and water + ninhydrin + [Cu(II)-Trp]$^+$) are existing down.

(a) [16-6-16]: 0.043 and 0.039 mmol kg$^{-1}$ at 303 K; 0.058 and 0.049 mmol kg$^{-1}$ at 353 K.
(b) [16-5-16]: 0.034 and 0.030 mmol kg$^{-1}$ at 303 k; 0.055 and 0.043 mmol.kg$^{-1}$ at 353 K.
(c) [16-4-16]: 0.032 and 0.025 mmol kg$^{-1}$ at 303 K; 0.043 and 0.033 mmol kg$^{-1}$ at 353 K.

## 2.3. Spectra of product formed

Spectra were obtained in aqueous system as well as gemini micellar system. Single-beam Shimadzu model spectroscope (UV mini 1240, Kyoto, Japan) was used to note the absorbance at different wavelengths ranged from 340 to 620 nm. Absorbance of product was drawn against varying wavelength and demonstrated graphically in figure 1. Absorbance values are developed more in surfactant system compared with aqueous system with unaffected absorption maximum (=370 nm).

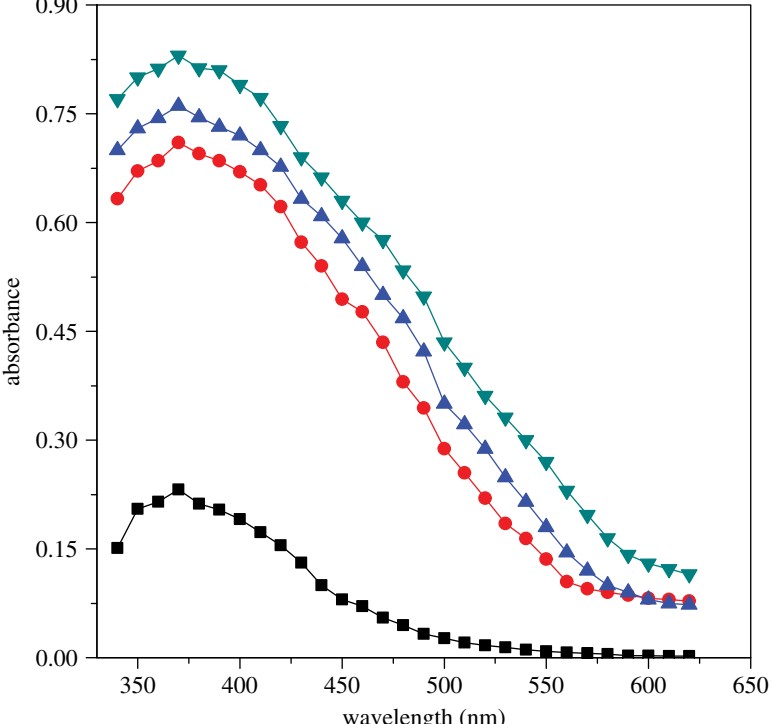

**Figure 1.** Spectra obtained in aqueous system as well as gemini micellar system on [Cu(II)-Trp]$^+$ and ninhydrin reaction at 353 K after heating 2 h: (rectangle) aqueous, (circle) 16-6-16, (triangle) 16-5-16 and (inverted triangle) 16-4-16. Experimental conditions: [ninhydrin] = 6 mmol kg$^{-1}$, [Cu(II)-Trp]$^+$ = 0.2 mmol kg$^{-1}$, [16-s-16] = 30 × 10$^{-2}$ mmol kg$^{-1}$ and pH = 5.0.

These results can be seen visually in figure 1. Consequently, figure 1 confirms that product formation is same in the two systems.

## 2.4. Kinetic measurements

In this study, the experiments were made under pseudo-first-order reaction circumstances fixing concentration of ninhdyrin in excess compared with complex concentration. Requisite volumes of gemini surfactant, acetate buffer, metal salt and amino acid were placed in a round-bottomed three-necked reaction pot. The pot was fixed in thermostated water bath at desired experimental temperature. The solution was left 30 min to ensure equilibrium. Kinetic experiments were performed by pouring a known volume of ninhydrin into the pot. So, the kinetic data were acquired under pseudo-first-order reaction circumstances at regular time intervals on UV–visible spectroscopy with identical quartz cuvettes of path length 1 cm. The rate constant ($k_\psi$) values were estimated as an average of at least triplicate runs. A detailed procedure in regard of kinetic measurements is available in the literature published previously [57–63].

## 2.5. Job's method

Job's method was used to inspect composition of product prepared on interaction of metal-amino acid complex and ninhydrin by heating complex and ninhydrin at 368 K for 2 h. Subsequently, absorbance was noted at $\lambda_{max}$ (= 370 nm) at the end by the means of UV–visible spectrophotometer (figure 2). It was observed that ninhydrin (1 mol) reacted with complex (1 mol) to yield the product.

# 3. Results

## 3.1. Influence of pH on $k_\psi$

Interaction of ninhydrin with metal amino acid complex at different pH was studied in the presence of gemini dicationic surfactants, keeping other parameters constant. The resultant values of rate constant obtained at different pH are mentioned in table 2. Also, rate constants are plotted at varying pH and

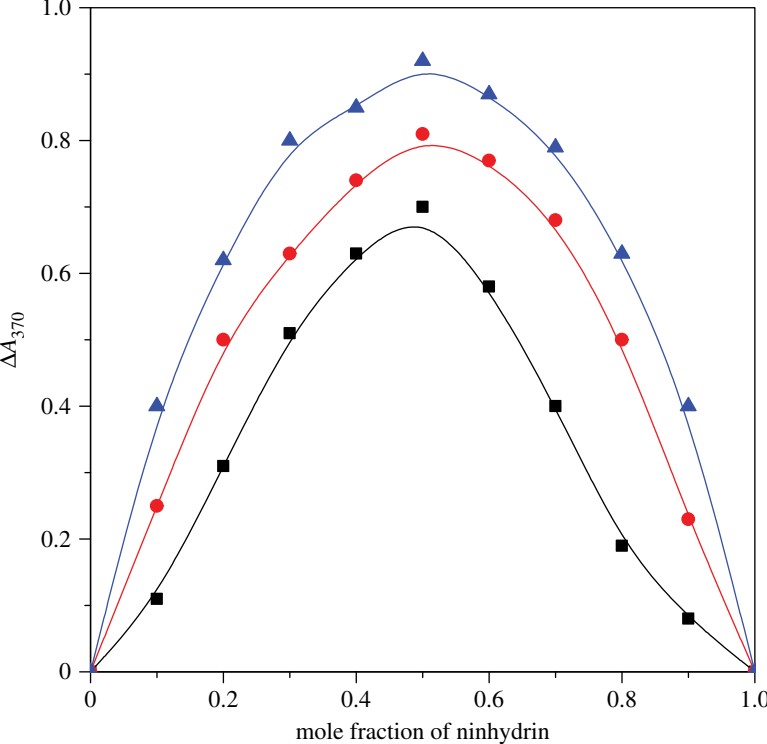

**Figure 2.** Plots of $A_{370}$ versus mole fraction of ninhydrin for estimation of product composition prepared on [Cu(II)-Trp]$^+$ and ninhydrin reaction by heating complex and ninhydrin at temperature 368 K for 2 h: (rectangle) 16-6-16, (circle) 16-5-16 and (triangle) 16-4-16. Experimental conditions: [16-$s$-16] $= 30 \times 10^{-2}$ mmol kg$^{-1.}$

**Table 2.** Implications of different factors on $k_\psi$ on [Cu(II)-Trp]$^+$ and ninhydrin reaction in geminis ($30 \times 10^{-2}$ mmol kg$^{-1}$) at [ninhydrin] (6 mmol kg$^{-1}$). Standard uncertainties are in $k_\psi = \pm 0.1 \times 10^{-5}$ s$^{-1}$.

| [Cu(II)-Trp]$^+$ (mmol kg$^{-1}$) | pH | temp. (K) | $10^5\ k_\psi$ (s$^{-1}$) | | |
|---|---|---|---|---|---|
| | | | 16-6-16 | 16-5-16 | 16-4-16 |
| 0.1 | 5.0 | 353 | 9.4 | 10.4 | 12.2 |
| 0.15 | 5.0 | 353 | 9.4 | 10.5 | 12.0 |
| 0.2 | 5.0 | 353 | 9.5 | 10.5 | 12.0 |
| 0.25 | 5.0 | 353 | 9.6 | 10.5 | 12.1 |
| 0.3 | 5.0 | 353 | 9.5 | 10.4 | 12.2 |
| 0.2 | 4.0 | 353 | 5.5 | 6.1 | 6.6 |
| 0.2 | 4.5 | 353 | 6.2 | 7.2 | 8.5 |
| 0.2 | 5.0 | 353 | 9.5 | 10.5 | 12.0 |
| 0.2 | 5.5 | 353 | 11.0 | 11.8 | 13.0 |
| 0.2 | 6.0 | 353 | 11.4 | 12.1 | 13.2 |
| 0.2 | 5.0 | 343 | 5.1 | 6.8 | 7.7 |
| 0.2 | 5.0 | 348 | 7.0 | 8.5 | 9.4 |
| 0.2 | 5.0 | 353 | 9.5 | 10.5 | 12.0 |
| 0.2 | 5.0 | 358 | 11.0 | 12.8 | 14.2 |
| 0.2 | 5.0 | 363 | 13.2 | 15.6 | 17.8 |

shown graphically in figure 3. Figure 3 reveals that rate increases up to pH 5, then becomes approximately constant. This behaviour confirms formation of Schiff base in vicinity of pH 5. As a consequence, studies were made at pH 5.

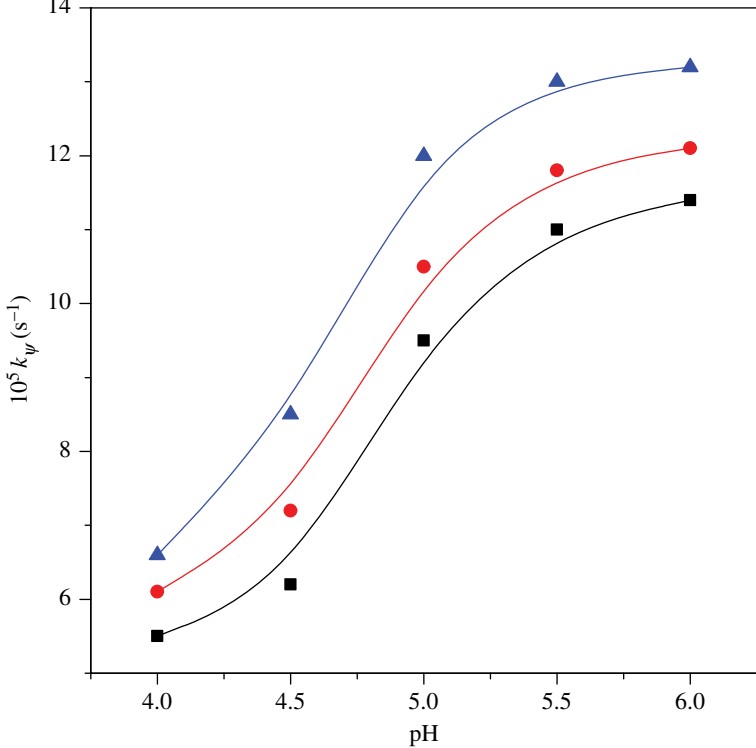

**Figure 3.** Implication of varying pH on $k_\psi$ for [Cu(II)-Trp]$^+$ and ninhydrin reaction in 16-s-16 surfactants: (rectangle) 16-6-16, (circle) 16-5-16 and (triangle) 16-4-16. Reaction conditions: [Cu(II)-Trp]$^+$ = 0.2 mmol kg$^{-1}$, [ninhydrin] = 6 mmol kg$^{-1}$, [16-s-16] = 30 × 10$^{-2}$ mmol kg$^{-1}$ and temperature = 353 K.

## 3.2. Influence of metal amino acid concentration on $k_\psi$

To check role of concentration of metal amino acid complex on rate constant, experiments were run at several initial concentrations of complex under pseudo-first-order reaction condition by fixing other experimental ingredients constant. The varied range of complex concentration was 0.1–0.3 mmol kg$^{-1}$. The evaluated values of $k_\psi$ at different initial complex concentrations are given in table 2. Evaluated results of table 2 confirmed that the study suggested a first-order dependence of rate in [complex]. Then rate equation can be expressed as equation (3.1).

$$\frac{\text{d[product]}}{\text{d}t} = \text{rate constant } (k_\psi) \times [M-AA]^+, \tag{3.1}$$

where [M-AA]$^+$ refers to [Cu(II)-Trp]$^+$.

## 3.3. Influence of ninhydrin concentration on $k_\psi$

Influence of ninhydrin concentration was carried out by varying ninhydrin ranging from 0 to 40 mmol kg$^{-1}$ in gemini micellar condition at fixed [complex], temperature and pH. Rate constant increases on increasing ninhydrin concentration. Rate values are plotted against several ninhydrin concentrations (figure 4). Plot of rate constant versus [ninhydrin] clearly demonstrates a nonlinear curve crossing through origin. This confirms order to be fractional in ninhydrin concentration.

## 3.4. Influence of temperature on $k_\psi$

In order to see the behaviour of temperature on the study, kinetic runs were made at five different temperatures, *viz.*, 343, 348, 353, 358 and 363 K at fixed concentration of reactants (ninhydrin and metal-amino acid) and pH in gemini surfactant system. The outcome of rates noted in the study are presented in tabular form in table 2. Rates increase with increasing temperature. Thermodynamic quantities such as $\Delta H^\#$, $\Delta S^\#$ and $E_a$ have been determined using Eyring equation in geminis. These thermodynamic quantities are reported in table 3.

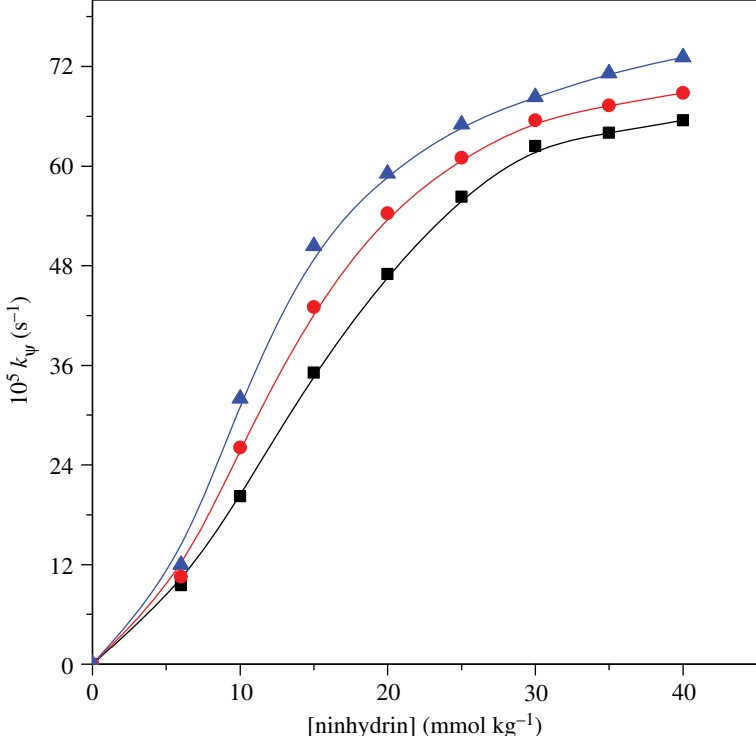

**Figure 4.** Implication of varying ninhydrin on $k_\psi$ for [Cu(II)-Trp]$^+$ and ninhydrin reaction in 16-s-16 surfactants: (rectangle) 16-6-16, (circle) 16-5-16 and (triangle) 16-4-16. Experimental conditions: [Cu(II)-Trp]$^+$ = 0.2 mmol kg$^{-1}$, [16-s-16] = 30 × 10$^{-2}$ mmol kg$^{-1}$, temperature = 353 K and pH = 5.0.

**Table 3.** Thermodynamic quantities ($E_a$, $\Delta H^\#$ and $\Delta S^\#$), $k_m$ and $K_E$ and $K_F$ calculated on ninhydrin (6 mmol kg$^{-1}$) and [Cu(II)-Trp]$^+$ (0.2 mmol kg$^{-1}$) reaction in geminis (30 × 10$^{-2}$ mmol kg$^{-1}$).

|  | aqueous[a] | 16-6-16 | 16-5-16 | 16-4-16 |
|---|---|---|---|---|
| $E_a$ (kJ mol$^{-1}$) | 60.5 | 33.8 | 32.3 | 30.1 |
| $\Delta H^\#$ (kJ mol$^{-1}$) | 57.7 | 31.0 | 29.5 | 27.3 |
| $-\Delta S^\#$ (JK$^{-1}$ mol$^{-1}$) | 143.7 | 170.4 | 171.5 | 172.9 |
| $10^3$ km (s$^{-1}$)[a] | — | 3.0 | 3.4 | 3.9 |
| $K_E$ (mol$^{-1}$ dm$^3$)[a] | — | 60.0 | 57.0 | 52.0 |
| $K_F$ (mol$^{-1}$ dm$^3$)[a] | — | 54.0 | 52.0 | 49.0 |

[a]At 353 K. Standard uncertainties are: $E_a$ = ±0.1 kJ mol$^{-1}$, $\Delta H^\#$ = ±0.1 kJ mol$^{-1}$ and $\Delta S^\#$ = ±0.1 J K$^{-1}$ mol$^{-1}$.

# 4. Discussion

## 4.1. Reaction mechanism

The proposed reaction mechanism of present study between ninhydrin and metal amino acid complex is shown as scheme 1. This is familiar previously that lone pair of nitrogen of amino group is mandatory for attack on middle carbonyl group of ninhydrin [64–67]. But, electrons of lone pair are connected to metal ion. Under such reaction condition, ninhydrin forms a complex with metal-amino acid. This is known as characteristic of combination-of-two-ligands-attached-to-the-same-metal-ion (CLAM) reaction mechanism [68–71].

## 4.2. Influence of gemini dicationic surfactants on the study

To determine the influence of geminis on the study, rate constants were calculated at several amounts of gemini surfactant concentration keeping other reaction factors fixed. These values of rate constant are summarized in electronic supplementary material, table S1.

**Scheme 1.** Reaction mechanism of present study between ninhydrin and [Cu(II)-Trp]$^+$. $K$ and $k$ stand for equilibrium and rate constants, respectively.

Rate constant increases steadily with increasing gemini at concentration below cmc value (Zone I) and levelling-off zones achieve at gemini concentration up to $400 \times 10^{-2}$ mmol kg$^{-1}$ (Zone II). Plots of Zone I and Zone II, figure 5, are detected the same as conventional monomeric surfactants [72–74]. At later stage, geminis result in a Zone III of increasing rate at higher surfactant concentration. Results suggested that the similar kinetics of rate with respect to ninhdyrin and metal-amino acid complex, i.e. fractional and first-order, respectively, were attained in gemini micellar medium as that to pure water medium.

## 4.3. Quantitative analysis of rate constant against gemini surfactants plot

Quantitative analysis of enhanced rate constant against [gemini] in the study can be interpreted successfully with model led by Menger & Portnoy [75] and established by Bunton [76,77].

In current study, the model is shown as scheme 2 below.

Equation (3.1) and scheme 2 gave equation (4.1)

$$k_\psi = \frac{k'_W + k'_m K_E[D_n]}{1 + K_E[D_n]}. \tag{4.1}$$

Then, equation (4.1) can be converted to equation (4.2)

$$k_\Psi = \frac{k_w[\mathrm{Nin}] + (K_E k_m - k_w)M_N^S[D_n]}{1 + K_E[D_n]}, \tag{4.2}$$

where $k_w$ and $k_\psi$ denote rate constants in pure water and gemini surfactants, respectively. $K_E$ and $K_F$ specify the respective binding constant of M-AA complex to micelle and ninhydrin to micelle. $M_N^S = [(\mathrm{Nin})_m]/[D_n]$ is concentration of ninhydrin in molar ratio of the micellar head group.

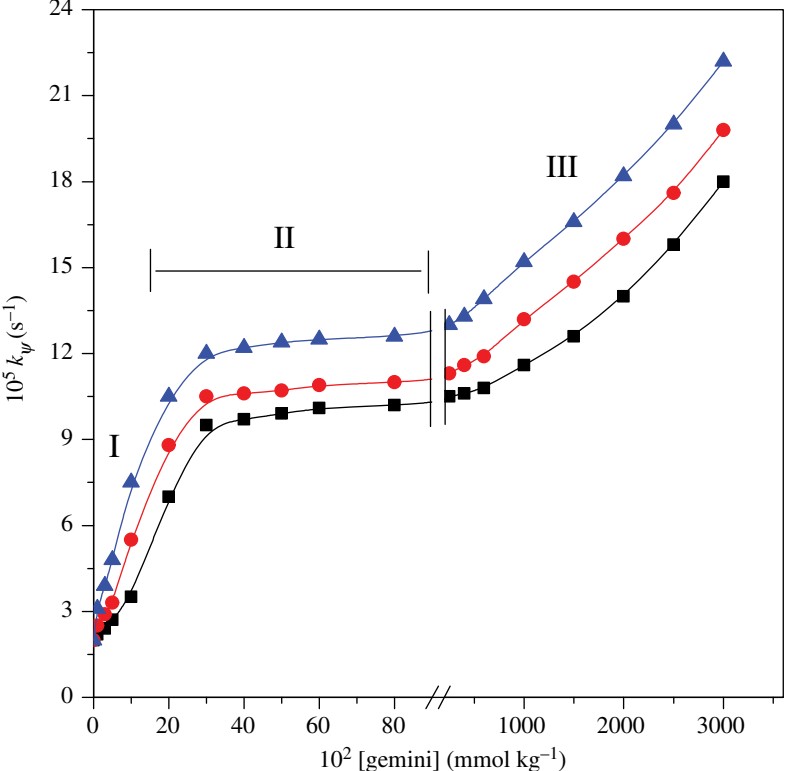

**Figure 5.** Implication of varying [gemini] on $k_\psi$ for [Cu(II)-Trp]$^+$ and ninhydrin reaction: (rectangle) 16-6-16, (circle) 16-5-16, (triangle) 16-4-16. Experimental conditions: [ninhydrin] = 6 mmol kg$^{-1}$, [Cu(II)-Trp]$^+$ = 0.2 mmol kg$^{-1}$, temperature = 353 K and pH = 5.0.

$$(\text{M-AA]}^+)_w \quad + \quad D_n \quad \underset{\longleftarrow}{\overset{K_E}{\longrightarrow}} \quad (\text{M-AA]}^+)_m$$

$$+ \qquad\qquad\qquad\qquad\qquad +$$

$$(\text{Nin})_w \quad + \quad D_n \quad \underset{\longleftarrow}{\overset{K_F}{\longrightarrow}} \quad (\text{Nin})_m$$

$$k'_w \qquad\qquad\qquad\qquad\qquad k'_m$$

$$\longrightarrow \text{product (P)} \longleftarrow$$

**Scheme 2.** Metal-amino acid and ninhydrin reaction in pure water and gemini micellar systems. $k_w$ ($= k'_w/[(\text{Nin})_w]$) and $k_m(= k'_m/M_N^S)$ refer to second rate constant in pure water and geminis. $D_n$ (= [total surfactant]-cmc) signifies micellized surfactant.

In order to get $K_E$ and micellar rate constant ($k_m$), we need cmc values under existing kinetic study. So, cmc values have been determined by the means of conductometric technique. For known cmc, $K_E$ and $k_m$ were calculated from equation (4.2) by a computer process. The values of $K_E$ and $k_m$ are provided in table 3. Putting of $K_E$ and $k_m$ in equation (4.2) results in the calculated $k_{\psi\text{cal}}$ which is in consistent with the observed $k_\psi$ (electronic supplementary material table S1). Electronic supplementary material, table S1 confirms the good matching between the observed $k_\psi$ and calculated $k_{\psi\text{cal}}$ within experimental errors, authenticating the proposed mechanism of present study.

Considering the consequences of Zone I (figure 5), [geminis] are lower than their cmc, $k_\psi$-values should be remained constant. Rate profile of $k_\psi$ versus [gemini] (figure 5) has confirmed an increment in rate constant. This may be owing to existence of premicellar aggregates between substrate and surfactant molecules even though at surfactant concentrations lower than that of their cmc values. It is approved well that gemini surfactants can form various morphological aggregates, such as vesicles, micelles and bilayers with different additives. It has also been noted that the surfactant molecules with substrate molecules formed premicellar aggregates and catalysed the reaction even at concentration lesser than cmc value [78–81].

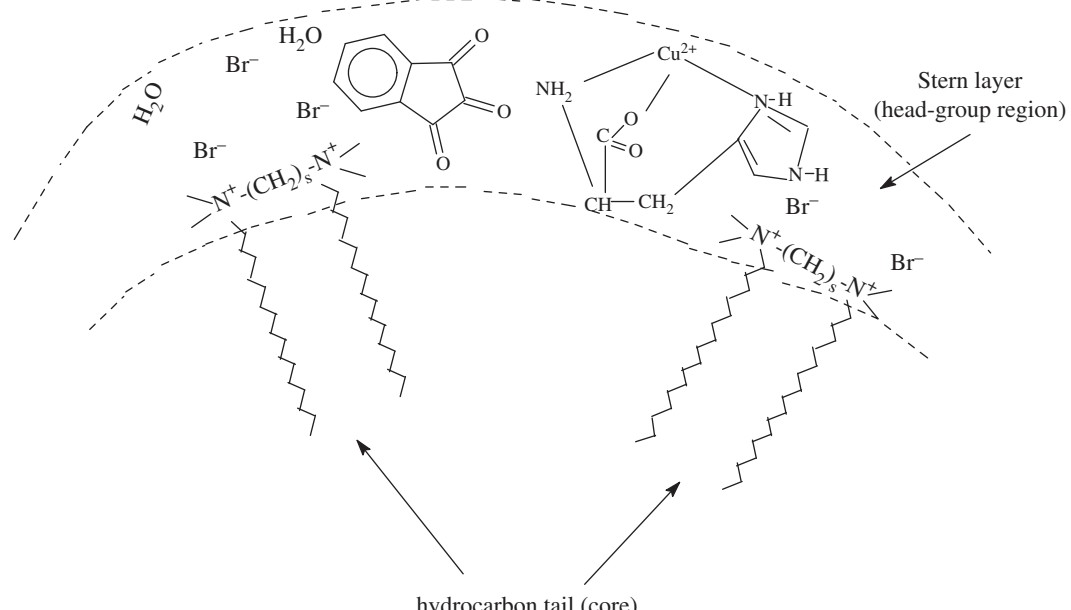

**Scheme 3.** Micellar structure and probable location of reactants in gemini micellar system. Spacer $(s)$ = 4, 5, 6.

According to multiple equilibrium models, the partition of surfactant between different states is governed by a several dynamic association and dissociation equilibrium. The smaller aggregates, such as dimer, trimer, tetramer etc. can be present at the concentration of surfactants below their cmc values

$$D + D \rightleftharpoons D_2 + D \rightleftharpoons D_3 \cdots D_{n-1} + D \rightleftharpoons D_n.$$

Rate constants become almost constant in Zone II. This happens when reactants are totally micellar bound with the micellar structure reflected to persist intact [82]. It was found that gemini surfactants were more functioning to catalyse the reaction than corresponding monomeric surfactant (cetyltrimethylammonium bromide, CTAB). This was the benefit of gemini surfactants used in the present kinetic study.

Gemini surfactants result in a Zone III of increasing rate at higher surfactant concentrations. Enhancement in rate occurs at higher gemini concentrations caused by changes in micellar structure and are a good match to $^1$H NMR spectral consequences stated previously [48,83]. Henceforward, an intensification in rate constant $k_\psi$ at higher surfactant concentrations follows as a result of modifications in morphological aggregates that delivers different experimental microenvironment, i.e. less polar.

All categories of micellar-mediated organic reactions (ionic, polar and neutral) are commonly believed to happen into small volume of a micelle (i.e. Stern layer) of an ionic surfactant.

Rate enhancement in positively charged micelles could be attributed to the stabilization of Schiff base intermediate on a positively charged micellar surface increasing the concentration of intermediate in the Stern layer. From electrostatic considerations, $\pi$-electrons existing in ninhydrin assist its possibility of partitioning between aqueous and positively charged micelles [84]. Hydrophobic interactions bring about incorporation of [Cu(II)-Trp]$^+$ into micelles. Therefore, both reactants ninhydrin and [Cu(II)-Trp]$^+$ get associated/incorporated into the aqueous surface of the micelles (i.e. the Stern layer) [76]. Therefore, the concentration of reactants increases into a small volume, that is, the Stern layer of the micelles (scheme 3), catalysing the reaction and resulting in an increase in the observed rate ($k_\psi$).

## 4.4. Thermodynamic quantities

Numerous thermodynamic quantities *viz.*, activation energy, $E_a$, activation enthalpy, $\Delta H^\#$, and activation entropy, $\Delta S^\#$, were evaluated on interaction of ninhydrin with metal amino acid in three gemini dicationic surfactant systems with Eyring equation. Obtained values of these thermodynamic quantities are listed in table 3. A lower value of activation enthalpy ($\Delta H^\#$) in gemini than the absence of surfactant (i.e. aqueous medium [85]) was obtained. This can be ascribed to the fact that an electrostatic attraction occurs between surfactants and reactant molecules when reactant molecules are existing in micellar phase. A reduced

value of activation entropy ($\Delta S^{\#}$) in gemini surfactants with those acquired in aqueous system confirms that the activated complex formed are well order in gemini surfactants.

## 5. Conclusion

In this present article, three gemini dicationic surfactants were synthesized and characterized consisting of two heads and tails connected covalently through a spacer by $^1$H NMR technique. The implications of their micellar solution on the study were performed with UV–visible spectroscopy. Studies were made at different experimental situations, e.g. temperature, pH, reactants and surfactant concentration. The cmc determination of gemini surfactants with and without additives was done on conductivity meter.

Under a set of varying experimental conditions, gemini micellar systems (even though at gemini surfactant concentrations lower than their cmc) were detected more effective to catalyse and accelerate the reaction over aqueous system. This suggested that the gemini surfactants were proved better surface active materials for the selected study. All the three gemini surfactants showed the order of their catalysing efficacies at each concentration as 16-4-16 > 16-5-16 > 16-6-16. Use of fairly small amounts of synthesized gemini surfactants in the study provides less environmental effect and reduces the catalytic competitions required as a catalyst in several industries. We trust that the specific outcomes of this study simplify an improved understanding of the reaction between ninhydrin and amine functional group. Study may reveal a new platform in intensifying the immense scope of uses of these gemini surfactants for scientific community in future.

Data accessibility. Data that supporting this study have been uploaded as electronic supplementary material.
Authors' contributions. D.K. has done the experiments and written the manuscript. M.A.R. and A.M.A. analysed and interpreted data. All authors gave final approval for publication.
Competing interests. The authors declare no competing interest.
Funding. We received no funding for this study.
Acknowledgements. Division of Computational Physics, Institute for Computational Science, Ton Duc Thang University, Ho Chi Minh City, Vietnam are highly acknowledged.

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
