## [Reviewer comments · Royal Society Open Science]

Review History

RSOS-200775.R0 (Original submission)

Review form: Reviewer 1

Is the manuscript scientifically sound in its present form?

Yes

Are the interpretations and conclusions justified by the results?

Yes

Is the language acceptable?

Yes

Do you have any ethical concerns with this paper?

No

Have you any concerns about statistical analyses in this paper?

No

Recommendation?

Accept with minor revision (please list in comments)

Comments to the Author(s)

The manuscript deals with the synthesis and characterization of geminis and implications of their micellar solution on ninhydrin and metal amino acid complex". This is a good work on the micellar catalysis. Authors have performed all experiments carefully and results are very clear. Although, I have certain corrections/suggestions as given below.

1. Before submitting the revised manuscript, pl. check all the abbreviations and symbols are well defined. In addition, check the typographical errors and correct them.
2. Some information about the instruments used are missed. Manuscript requires sufficient information about all the instruments employed during the study.
3. In section 2.2, what were the concentrations of ninhydrin and complex used for the cmc measurements of gemini surfactants?
4. How many times were each analysis carried out? Mention in the revised manuscript.
5. Results and Discussions should be separated. This will also make reader understand better as it contains many sections. It will be good to make subsections under a new section (Discussions).
6. Check the rate equation (1) mentioned in section 3.2 and try to generalize it.
7. Implication of ninhydrin on the reaction provided in Table 2 should be deleted as the same is plotted in Fig. 4.
8. Authors should check the references and try to update them wherever it is applicable/possible.

Review form: Reviewer 2

Is the manuscript scientifically sound in its present form?

Yes

Are the interpretations and conclusions justified by the results?

No

Is the language acceptable?

Yes

Do you have any ethical concerns with this paper?

No

Have you any concerns about statistical analyses in this paper?

No

Recommendation?

Major revision is needed (please make suggestions in comments)

Comments to the Author(s)

1. The titles of the sections sound somewhat strange: "3.1. pH"; "3.2. Metal amino acid concentration"; etc., etc.
2. The numeration of the sections is strange: "3. Results and discussion". "4. Reaction mechanism". "5. Influence of gemini dicationic surfactant on the study"; etc. Is it also a part of Results and discussion?
3. Page 12, Line 49: the phrase "... no reaction was occurred in Zone II" are misleading, because one can think that it concerns the traction of metal complex formation.
4. On the other hand, in Zone III the reaction rate increases, Figure 5. However, over CMC the rates in micellar media normally decrease owing to the dilution of reactions by the pseudophase.

5. What is the locus of the reactants in micelles?
6. Nothing is said about the Stern layer, surface charge of the micelles and the electrostatic potential.

Decision letter (RSOS-200775.R0)

Dear Dr Kumar:

Title: Synthesis and characterization of geminis and implications of their micellar solution on ninhydrin and metal amino acid complex
Manuscript ID: RSOS-200775

The editor assigned to your manuscript has now received comments from reviewers. We would like you to revise your paper in accordance with the referee and Subject Editor suggestions which can be found below (not including confidential reports to the Editor). Please note this decision does not guarantee eventual acceptance.

Please submit your revised paper before 21-Jun-2020. Please note that the revision deadline will expire at 00.00am on this date. If we do not hear from you within this time then it will be assumed that the paper has been withdrawn. In exceptional circumstances, extensions may be possible if agreed with the Editorial Office in advance. We do not allow multiple rounds of revision so we urge you to make every effort to fully address all of the comments at this stage. If deemed necessary by the Editors, your manuscript will be sent back to one or more of the original reviewers for assessment. If the original reviewers are not available we may invite new reviewers.

RSC Associate Editor:
Comments to the Author:
(There are no comments.)

RSC Subject Editor:
Comments to the Author:
(There are no comments.)

Reviewers' Comments to Author:
Reviewer: 1

Comments to the Author(s)

The manuscript deals with the synthesis and characterization of geminis and implications of their micellar solution on ninhydrin and metal amino acid complex". This is a good work on the micellar catalysis. Authors have performed all experiments carefully and results are very clear. Although, I have certain corrections/suggestions as given below.

1. Before submitting the revised manuscript, pl. check all the abbreviations and symbols are well defined. In addition, check the typographical errors and correct them.
2. Some information about the instruments used are missed. Manuscript requires sufficient information about all the instruments employed during the study.
3. In section 2.2, what were the concentrations of ninhydrin and complex used for the cmc measurements of gemini surfactants?
4. How many times were each analysis carried out? Mention in the revised manuscript.
5. Results and Discussions should be separated. This will also make reader understand better as it contains many sections. It will be good to make subsections under a new section (Discussions).
6. Check the rate equation (1) mentioned in section 3.2 and try to generalize it.
7. Implication of ninhydrin on the reaction provided in Table 2 should be deleted as the same is plotted in Fig. 4.
8. Authors should check the references and try to update them wherever it is applicable/possible.

Reviewer: 2

Comments to the Author(s)

1. The titles of the sections sound somewhat strange: "3.1. pH"; "3.2. Metal amino acid concentration"; etc., etc.
2. The numeration of the sections is strange: "3. Results and discussion". "4. Reaction mechanism". "5. "Influence of gemini dicationic surfactant on the study"; etc. Is it also a part of Results and discussion?

3. Page 12, Line 49: the phrase "... no reaction was occurred in Zone II" are misleading, because one can think that it concerns the traction of metal complex formation.
4. On the other hand, in Zone III the reaction rate increases, Figure 5. However, over CMC the rates in micellar media normally decrease owing to the dilution of reactions by the pseudophase.
5. What is the locus of the reactants in micelles?
6. Nothing is said about the Stern layer, surface charge of the micelles and the electrostatic potential.

Author's Response to Decision Letter for (RSOS-200775.R0)

See Appendix A.

RSOS-200775.R1 (Revision)

Review form: Reviewer 1

Is the manuscript scientifically sound in its present form?

Yes

Are the interpretations and conclusions justified by the results?

Yes

Is the language acceptable?

Yes

Do you have any ethical concerns with this paper?

No

Have you any concerns about statistical analyses in this paper?

No

Recommendation?

Accept as is

Comments to the Author(s)

All the comments have been removed and the manuscript can be accepted for publication in its present form.

Review form: Reviewer 2

Is the manuscript scientifically sound in its present form?

Yes

Are the interpretations and conclusions justified by the results?

Yes

Is the language acceptable?

Yes

Do you have any ethical concerns with this paper?

No

Have you any concerns about statistical analyses in this paper?

No

Recommendation?

Accept as is

Comments to the Author(s)

The improved version of the manuscript is publishable.

Decision letter (RSOS-200775.R1)

Dear Dr Kumar:

Title: Synthesis and characterization of geminis and implications of their micellar solution on ninhydrin and metal amino acid complex

Manuscript ID: RSOS-200775.R1

It is a pleasure to accept your manuscript in its current form for publication in Royal Society Open Science. The chemistry content of Royal Society Open Science is published in collaboration with the Royal Society of Chemistry.

RSC Associate Editor:
Comments to the Author:
(There are no comments.)

RSC Subject Editor:
Comments to the Author:
(There are no comments.)

Reviewer(s)' Comments to Author:
Reviewer: 1

Comments to the Author(s)
All the comments have been removed and the manuscript can be accepted for publication in its present form.

Reviewer: 2

Comments to the Author(s)
The improved version of the manuscript is publishable.

Appendix A

Journal Title: Royal Society Open Science

Manuscript Title: Synthesis and characterization of geminis and implications of their micellar solution on ninhydrin and metal amino acid complex

Manuscript ID: RSOS-200775

Dear Professor Smith,

Thank you very much for your useful comments. We have modified the manuscript in the light of Reviewers' comments. Detailed corrections are listed below point by point.

Thanking you,

Sincerely yours,

Dr. Dileep Kumar

Response to Reviewer # 1:

1. All abbreviations (Pl. see Abstract on Page 2) and symbols (Pl. see Abstract on Page 2 and Scheme 1 on Page 8) used are well defined. Typographical errors are removed (Pl. see Introduction on Page 3; Section 2.1 on Page 4; Section 2.4 on Page 6; Section 3.3 on Page 7; Section 4.2 on Page 8; Conclusions on Page 12 and Table 1 on Page 22).
2. Sufficient information about all the instruments employed are given (Pl. see line 3 on Page 5; Section 2.2 on Page 5 and Section 2.3 on Page 5).
3. Concentrations of ninhydrin and complex used are mentioned in Section 2.2 (Pl. see on Page 5).
4. Required information is provided (Pl. see Section 2.2 on Page 5 and Section 2.4 on Page 6).
5. Thank you for comments. Changes are made as per suggestion (Pl. see on Pages 6-9 and 12).
6. Now, equation (1) is modified (Pl. see on Page 7).
7. Data of ninhydrin are deleted from Table 2 (Pl. see on Page 23).

8. References are corrected and updated (Pl. see references [11], [14], [53], [64], [67], [69] & [83]).

Response to Reviewer# 2:

1. Thank you for your concern. Now, title of sections are modified (Pl. see on Pages 6 & 7).
2. Numeration of sections is now corrected (Pl. see on Pages 6-9 & 12).
3. Thank you very much for your advice. Now, sentence is rephrased (Pl. see on Page 10).
4. Thanks for your comments. In Zone III, reaction rate increases. This happens due to the presence of spacer in the geminis which decreases the water content in the aggregates making the environment less polar and thus causing rate increases [a]. Menger et al. [b] have already concluded that due to proximity of positive charges in gemini micelles anion binding at surfaces is increased at the expense of binding of H₂O.
 - (a) Kabir-ud-Din, Fatma W. 2007 *J. Phys. Org. Chem.* **20**, 440–447.
 - (b) Menger FM, Keiper JS, Mbadugha BNA, Caran KL, Romsted LS. 2000 *Langmuir* **16**, 9095–9098.
5. Locus of reactants in micelles is shown in Scheme 3 (Pl. see on Page 11).
6. Required description is provided (Pl. see on Pages 11 and 12).